# Chitosan Film Sensor for Ammonia Detection in Microdiffusion Analytical Devices

**DOI:** 10.3390/polym15214238

**Published:** 2023-10-27

**Authors:** Irene Tagliaro, Giacomo Musile, Paolo Caricato, Romolo M. Dorizzi, Franco Tagliaro, Carlo Antonini

**Affiliations:** 1Department of Materials Science, University of Milano, via Cozzi 55, 20131 Milano, Italy; carlo.antonini@unimib.it; 2Unit of Forensic Medicine, Department of Diagnostics and Public Health, University of Verona, Piazzale L. A. Scuro, 10, 37134 Verona, Italy; romolo.dorizzi@gmail.com (R.M.D.); franco.tagliaro@univr.it (F.T.); 3Directorate-General for Health and Food Safety G5, Food Hygiene, Feed and Fraud 03/104, 1049 Brussels, Belgium; paolo.caricato@ec.europa.eu

**Keywords:** chitosan, sensor, film, ammonia detection, Nessler’s reagent

## Abstract

Chitosan films have attracted increased attention in the field of sensors because of chitosan’s unique chemico-physical properties, including high adsorption capacity, filmability and transparency. A chitosan film sensor was developed through the dispersion of an ammonia specific reagent (Nessler’s reagent) into a chitosan film matrix. The chitosan film sensor was characterized to assess the film’s properties by Fourier transform infrared spectroscopy (FTIR), thermogravimetric analysis (TGA), scanning electron microscopy (SEM) and differential scanning calorimetry (DSC). A gas diffusion device was prepared with the chitosan film sensor, enabling the collection and detection of ammonia vapor from biological samples. The chitosan film sensor color change was correlated with the ammonia concentration in samples of human serum and artificial urine. This method enabled facile ammonia detection and concentration measurement, making the sensor useful not only in clinical laboratories, but also for point-of-care devices and wherever there is limited access to modern laboratory facilities.

## 1. Introduction

Chitosan-based materials have been developed for sensing applications in different analytical fields, including electrochemical biosensors [1,2], piezoelectric sensors [3] and pH indicators [4,5]. The extraordinary characteristics of chitosan, i.e., its adsorption, biodegradability, biocompatibility, filmability and transparency, make it a promising substrate for composite film preparation, compared to other synthetic polymers with a higher impact as raw materials [6]. Chitosan’s ability to adsorb or bond molecules and particles confers specific functionalities to materials [7]. Chitosan composites are easily filmed, because of the viscosity in solution, and allow color change detection by the naked eye. Due to its biodegradability and biocompatibility, chitosan is frequently used as an additive in food and biomedical devices [8].

Chitosan films have been loaded with organic and inorganic compounds for producing biopolymeric sensors, useful tools for onsite screening [9,10,11], semi-quantitative analysis [12,13,14] and intelligent packaging [15,16,17]. Especially in the field of food preservation and monitoring, different pH indicators have been loaded into chitosan-based matrices [18,19,20,21,22,23]. These devices were intended for food preservation or for quality monitoring through the development of responsive devices, using a biocompatible substrate, where a specific reaction/color change allows the detection of a specific marker [24,25,26,27,28].

Besides food applications, the determination of ammonia is crucial in other fields, especially in clinical chemistry [29,30]. Indeed, the increase of ammonia in biological fluids is associated with various disorders, particularly renal and liver diseases [31]. Ammonia is generated mostly from the action of urease on the breakdown of urea into ammonia and carbon dioxide. Through this reaction, colonic bacteria in the intestine generates ammonia from protein degradation, which enriches the blood stream. Through the portal vein, the blood enriched with ammonia flows to the liver, where urea, which is less toxic, is restored from ammonia. The liver converts 85% of the ammonia into urea, which can be excreted by the kidneys and the colon. The alteration of these processes is associated with severe health issues. The production of ammonia is increased in several conditions, for e.g., in convulsive seizures with increased muscle production, while, in the opposite case, ammonia levels are low because of the impairment of its clearance, for e.g., in hepatocellular dysfunction, portosystemic shunting, or both, with the subsequent impaired hepatic detoxification of ammonia [32,33,34]. For traditional detection methods, the determination of ammonia in biofluids is based on enzymatic analysis and ion-selective electrodes [35], with the former being the most common approach. In short, the method utilizes the reaction of glutamate dehydrogenase, where glutamate, NAD(P)+ and water results from the reaction between α-oxoglutarate and reduced nicotinic adenine dinucleotide (phosphate) (NAD(P)H) with ammonium [34]. Therefore, the decrease in the concentration of NADH or NADPH is measured by absorbance spectroscopy. A different direct method for detecting ammonia is the ammonium electrode, where a NH_4_^+^ selective membrane, based on a mixture of the antibiotics nonactin and monoactin, is used. Moreover, the actual analytical methods for ammonia determination lack easy, portable screening devices, which combine stability and affordable costs. Within this context, taking advantage of ammonia volatility at an alkaline pH, microdiffusion methods have also been proposed, particularly in point-of-care devices; however, due to the high costs, limited stability and complex operation, they have found only limited application [36].

For the determination of ammonia, an inorganic regent (Nessler’s reagent) is commonly used for water quality analysis, because of its specific reaction with ammonia. Nessler’s reagent is composed of a mixture of potassium tetraiodomercurate (II) and potassium hydroxide. When dissolved in water, the transparent solution turns instantaneously to yellow for a low amount of ammonia, and to deep red for a higher concentration of the analyte. Nessler’s reagent is very sensitive, as it can be used to detect quantities down to the µg of ammonia [37,38]. Potassium tetraiodomercurate (II) reacts with ammonia only at basic pHs and the intensity of the color is directly proportional to the ammonia concentration in the sample.

Therefore, in this study we combined chitosan properties as a transparent support film and Nesser’s reagent for its ammonia sensitivity, to develop and test an innovative analytical platform for rapid screening purposes. The chitosan film sensor (CFS) containing Nessler’s reagent was applied in a preliminary application to biological fluids.

## 2. Materials and Methods

### 2.1. Materials

All chemicals were of analytical reagent grade. Chitosan powder (DD 80%, MW 230,000, CAS 9012-76-4), Nessler′s reagent, glacial acetic acid (CAS 64-19-7), sodium hydroxide, ammonia 32%, human serum (no detectable ammonia) and the synthetic negative (no detectable ammonia) urine control were from Sigma-Aldrich (Darmstadt, Germany). A model Purelab1-Chorus 3 water purification system (Elga Veolia, High Wycombe, UK) was used to obtain ultra-pure water, to prepare all the solutions.

### 2.2. Methods

SEM analyses were performed with a Zeiss FEG Gemini 500 electronic microscope. The microscope can operate with accelerating voltages of 0.5–30 kV, beam currents of 3 pA-20 nA, with a nominal resolution of 0.6 nm at 15 kV. For the analysis reported in this study, the system was operating at 5 kV. The FEG–SEM was also equipped with an integrated Bruker QUANTAX EDS/WDS (energy dispersive/wave dispersive) microanalysis system for analysis of the light elements.

Attenuated total reflection–Fourier transform infrared spectroscopy (ATR-FTIR) was performed to acquire the infrared spectra, using a Thermo Fisher Scientific Nicolet iS20 FTIR spectrometer mounted with the Smart iTX accessory (4 cm^−1^ resolution, 650–4000 cm^−1^ range, 28 scans), equipped with a monolithic diamond crystal.

Thermogravimetric analysis (TGA) was performed using a Mettler Toledo TGA/DSC1 STAR e system. Weight variations were analyzed in a temperature range from 30 to 1000 °C, at 10 °C/min, under an air flux of 50 mL/min.

DSC analysis was performed using a Mettler Toledo DSC3 STAR e system. Samples were scanned in aluminum pans, under a nitrogen atmosphere, in the −70, 120 °C temperature range, with a heating/cooling rate of 5 °C/min (ramp: from 25 °C to 120 °C; 2 min at 120 °C; from 120 °C to −70 °C; 2 min at −70 °C; from −70 °C to 120 °C; 2 min at 120 °C; from 120 °C to 25 °C).

CFS reactivity was assessed, detecting the ammonia development from complex biofluids, i.e., serum and artificial urine. The change in color from transparent to red was associated with the effective reaction of ammonia with Nessler’s reagent. The test was performed by spiking samples with ammonia in the range from 28 µg/L to 28 g/L (28, 280, 2800 µg/L, 28, 280, 2800 mg/L and 28 g/L), in triplicate.

### 2.3. Preparation of Chitosan Film Sensor (CFS) for Ammonia Detection

The CFS for ammonia detection was prepared by simply dissolving chitosan and Nessler’s reagent and drying the solution by solvent casting. For preparing a film with a diameter of 5 cm, 50 mg of chitosan was dissolved in 5 mL of deionized water and 250 μL of acetic acid and stirred for 30 min. Nessler’s reagent, 0.75 mL, was then added to the solution. The mixture was subject to magnetic stirring for 90 min and then poured onto a glass Petri dish with a diameter of 5 cm. The Petri dish was placed in a ventilated oven at 30 °C overnight. The resultant film was peeled from the glass support and used for the ammonia detection.

A pure chitosan film was also prepared for a comparative evaluation and the same procedure was adopted. Chitosan (50 mg) was dissolved in 5 mL of deionized water and 250 μL of acetic acid and stirred for 2 h. The dissolved chitosan was poured into a Petri dish (5 cm diameter) and dried in a ventilated oven at 30 °C overnight.

### 2.4. Fabrication of the Gas Diffusion Device

The gas diffusion device consisted of a chromatography glass vial of 1.5 mL, with an insert of 0.4 µL, where the Teflon^®^ septum of the cap was replaced with a CFS with a proper size (0.9 cm, diameter of the CFS cap septum), as depicted in Figure 1. Ammonia evaporation is promoted by the basic pH solution; since the vial is a closed system, external disturbances to the liquid–gas interface are prevented. The device design resembled a previously described unit used for detecting ammonia in vitreous humor samples [39]. The method exploits a colorimetric reaction between molecular ammonia and the Nessler’s reagent (Equation (1)).
2[HgI_4_]^2−^ + 4OH^−^ + NH_4_^+^ → HgO · Hg(NH_2_)I + 7I^−^ + 3H_2_O(1)

Moreover, 200 μL of the sample was pipetted into the chromatography vial containing 30 mg of NaOH, the vial was then capped with the CFS allocated in the lid and 10 μL of a solution 6M of sodium hydroxide was dropped onto the CFS. The reaction and the vapor collection took place at room temperature. Color recording was performed with a smartphone camera (iPhone 13, Apple Inc., Cupertino, CA, US), using image acquisition by the device sensing part. The evaluation of the interaction between molecular ammonia and Nessler’s reagent was estimated using the ImageJ software version 1.8.0, using a deconvolution of the red, green and blue components, to obtain the *RGB* distance (Δ*RGB*), using Equation (2):(2)$$∆RGB=\sqrt{\left(R-R_{0}\right)+\left(G-G_{0}\right)+\left(B-B_{0}\right)}$$

The values *R*_0_, *G*_0_ and *B*_0_ (Equation (2)) correspond to the *R*, *G* and *B* components of a sample solution, which is analyzed for each matrix, and it is considered as the reference value. The analysis of a reference sample, i.e., biofluid without the analyte, minimizes the differences due to external light exposition or to the light camera-embedded smartphone. Also, the analysis of a reference sample reduces the differences due to the de-convolution performance related to the digitalization of the image, operated by the device used for recording the image.

The gas diffusion device testing setup is extremely easy to use for on-site analysis, since it requires only adding the sample inside the gas diffusion device (Figure 1(aii)) and a drop of NaOH 6M on the CFS lid. The color detection is conveniently conducted through a phone camera. This method obviously implies that the gas diffusion devices have been prepared in a laboratory, with NaOH inside the vial and the CFS placed inside the hollow lid.

## 3. Results and Discussion

### 3.1. Chitosan Film Sensor (CFS) Containing Nessler’s Reagent

A film sensor for ammonia analysis needs to be transparent for the easy evaluation of color changes (Figure 2b). Other characteristics for the CFS preparation are: (i) reduction to a minimum of chitosan used, and (ii) optimization of the Nessler’s reagent load for achieving the highest sensitivity to ammonia.

Since chitosan is soluble only in mildly acidic aqueous solutions, the direct dissolution of chitosan in the Nessler’s reagent is not possible, because of the presence of KOH that confers a strongly alkaline pH. Therefore, our procedure involves a first step on the dissolution of chitosan in an acidic aqueous solution, where the chitosan polymeric chains are protonated due to the presence of amine functional groups. When chitosan is dissolved, the Nessler’s reagent can be added to the chitosan–acetic acid solution with a consequent pH increase from 3.0 to 4.5. This step allows chitosan to remain protonated, thus maintaining its solubility. The Nessler’s reagent addition causes a slight change in the solution color from perfectly transparent to slightly yellowish opaque. In about one and a half minutes, the mixture again becomes transparent. This can be explained by hypothesizing that the coordination of HgI_4_^2−^ and K^+^ ions with the chitosan functional groups (i.e., R-OH) occurs during the mixing, resulting in a homogeneous ion distribution inside the dissolved chitosan matrix. During the drying process, Nessler’s components are trapped inside the chitosan structure resulting in a thin transparent film, which can easily be detached from the casting container and used for this purpose (Figure 2a).

The amount of chitosan was chosen with the aim of producing a thin self-standing transparent film, where the surface color changes due to the reaction with ammonia can be easily identified. For this purpose, 50 mg of chitosan proved enough to obtain a film with an area of about ~20 cm^2^ (i.e., the surface of a Petri dish with a diameter of 5 cm) and with a film thickness of 15 μm (±3) (Appendix A). This thickness allowed the highly transparent film’s easy detachment from the casting container.

Different amounts of Nessler’s reagent have been added to chitosan to obtain a color change of the CFS in the presence of ammonia, with an optimum at K_2_HgI_4_ 20% of the total weight of the CFS. These preliminary tests were conducted varying the K_2_HgI_4_ % from 15% to 30%. Although chitosan showed very high adsorption of the Nessler’s reagent (2.0 up to 4.5 times of its weight, calculated over the total dry content in the Nessler’s reagent), if the addition of Nessler’s reagent produces a pH exceeding 5–6, chitosan loses its solubility, resulting in non-homogeneous solutions and non-transparent films. Therefore, the optimum amount of K_2_HgI_4_ was established at 20% in weight, taking into consideration the detection of ammonia (appreciated change in color of the CFS) and the solubility of chitosan in the solution.

### 3.2. Physico-Chemical Characterization of the CFS

CFSs have been analyzed by SEM to assess the morphology. Figure 3a,c shows a comparison between a pure chitosan film and a CFS. The pure chitosan film (Figure 3a) has a homogeneous surface, where the profiles of solvent evaporation can be observed. These patterns are typical for solvent casting of polysaccharide films. The CFS (Figure 3c) also reveals profiles of evaporation, along which crystalline salt residues are clearly visible. Analyzing the same samples using EDX, information on the distribution of Nessler’s reagent inside the chitosan matrix was achieved (Figure 3b,d). While the pure chitosan film obviously shows only carbon and oxygen atoms (Figure 3b), the CFS reveals the presence of Hg, I and K (Figure 3d). Appendix A shows the distribution of the individual elements, Hg, I and K, which further confirms that a composite material was obtained. Using EDX analysis, over the analyzed area, the homogeneous distribution of Hg, I and K atoms was observed. The addition of Nessler’s salt obviously affects the composition and decreases the number of solvent evaporation profiles visible per area. The presence of salt residues does significantly impact the film’s transparency.

Using FTIR analysis, it was possible to investigate the interaction between chitosan and other ions, comparing the pure chitosan film spectra to that of the CFS (Figure 4a), for assessing the formation of weak and ionic bonds. The FTIR spectrum of the pure chitosan film showed the characteristic peaks of chitosan films dissolved in acetic acid solutions [40]. Between 3600 and 3000 cm^−1^, the broad absorption band was correlated to the stretching of hydroxyl and amino groups in the carbohydrate ring. Moreover, 1642 cm^−1^ is assigned to the stretching vibration of the carboxyl group (C=O, amide I), while the peaks at 1545 cm^−1^ are assigned to the bending of the N-H, amide II [41]. The FTIR vibration wavenumber of 1545 cm^−1^ and 1405 cm^−1^ were contemporaneously influenced by asymmetric and symmetric stretching of the carboxylate ion, ascribed to the vibrations by the acetate ion in the chitosan acetate salt [42].

The FTIR spectrum of the CFS showed an increase in the intensity in the OH stretching at ~3300 cm^−1^, due to the addition of KOH. The region interested in the amide vibrations showed peaks with increased intensity, which slightly shifts to 1648 cm^−1^ (C=O, amide I) and 1550 cm^−1^ (N-H, amide II), wavenumbers that are also interested in the vibrations by the acetate [43,44]. These changes may suggest differences in the chitosan coordination due to the additions of Nessler’s reagent ions and/or may result from the deprotonation of NH_3_^+^ by OH^−^ ions.

The TGA degradation profile of CFS (see Figure 4b) from room temperature to 1000 °C was compared with a pure chitosan film and a sample of Nessler’s reagent powder, which was previously desiccated in the oven. All the curves show a weight loss until 150 °C, associated with water desorption. Comparing the CFS with the pure chitosan film, we noticed that the CFS had a higher water content with a weight loss of ~20%, and only 10% for pure chitosan [41]. The presence of Nessler’s reagent (weight loss 13% to 150 °C), which is a hygroscopic solid, especially because of the presence of KOH, increases the water content. This aspect was also confirmed by the FTIR analysis. Around 200 °C, all the curves showed sharp weight losses. Pure chitosan undergoes degradation according to a two-step process, reaching complete degradation at around 800 °C [45] (weight loss 96%). The CFS showed a different degradation process, characterized by three major regimes (weight losses: 34%, 44%, 77%). At 1000 °C, CFS does not completely degrade due to Nessler’s salt causing incomplete degradation, as higher temperatures would be required.

Figure 4c shows a comparison between the DSC curves for the CFS and the pure chitosan film. A first endothermic peak is visible during the first heating ramp and was associated with the evaporation of absorbed water in the polymer. For the pure chitosan film, the peak is centered at 65 °C, while for the CFS we observed a shift at 105 °C. This shift is attributed to the presence of Nessler’s reagent, which is able to bind an increased amount of water, as also confirmed by the TGA analysis. The endothermic peak disappears in the second heating ramp because of water evaporation. No clear T_g_ transition is visible for any of the two materials, although other studies have reported a T_g_ of 54 °C for films containing acetic acid [40]. In this range of DSC temperature investigation, the thermal properties of pure chitosan do not significantly differ from CFS.

### 3.3. Gas Diffusion Device Application for Ammonia Detection

The CFS was allocated in the lid of a chromatography vial, creating a gas diffusion device where the volatile ammonia liberated in the head space is in equilibrium with the ammonia/ammonium ion present in the liquid phase. When NaOH is added, the equilibrium is forced from the ammonium ion to ammonia, which diffuses in the gas phase. As such, ammonia interacts with Nessler’s reagent in the CFS and produces a color change, which is recorded after 180 s (from the addition of NaOH in the sample) and evaluated by image analysis. The linearity was tested in the range concentration between 28 µg/L to 28 g/L in different matrices, i.e., serum and artificial urine.

For the spiked serum samples (Figure 5a), a linear trend of the log (C) is observed for the entire tested range of the concentration, with a correlation coefficient (R^2^) equal to 0.9609. The relationship between the *RGB* (y) and the log (ammonium concentration) (x) is expressed in the caption for Figure 5. For the spiked artificial urine samples, two ranges where identified, where the linearity of the function can be applied: 28 µg/L–28 mg/L, and 28 mg/L–28 g/L (Figure 5b). The former linear interval shows a correlation coefficient (R^2^) equal to 0.8411 and the latter equal to 0.9797.

The two slopes reflect the different capability of the device to turn to a different color. In particular, a slope of 5.9 (caption Figure 5) indicated a lower capability of the CFS to react with ammonia produced from low concentrated urine samples, while the chitosan-based film was more sensitive (slope equal to 37.1) to color changes if the concentration of ammonium in the urine samples was in the order of milligrams per liter. Differently, the sensitivity to ammonia produced from the serum samples seemed to show a constant capability to change color for the investigated range.

The developed device has been designed to simplify the determination of ammonia at the point of need; as such, the performance in terms of the limit of detection has been evaluated as the concentration causing a color change that can be identified by the operator. The visual color change for the spiked serum samples was identified at 0.28 mg/L. The limit of detection (LOD) with the naked eye for urine analyses was 28 mg/L.

Based on the above results, the developed device appears suitable for detecting ammonia in serum and urine samples for pathological conditions. The ammonia detoxification process is conducted in the organism through the conversion in urea. An alteration of this process corresponds to increases of ammonia in the organism. The present strategies in preliminary analysis demonstrate the feasibility to detect a severe increase of ammonia in biofluids, such as serum and urine, reflecting a pathological status. Indeed, the normal content in serum samples is within the range 0.2–1 mg/L [46], and the concentration of ammonium in urine samples is 0.1–1 g/L [47], as these values are well above the experimentally measured LOD.

Currently, the determination of ammonia in biofluids, such as serum and urine, is conducted by enzymatic-based reactions and ion-selective electrodes. Although these methods are the most diffused approaches offering extensively validated techniques, in comparison to the proposed approach, the current methods are more expensive, and in most cases are designed to be used in laboratories, instead of at the point of need. Also, another drawback to the enzymatic method is the time of analysis, which is 90 min [48]. On the other hand, the present method showed higher limits of detection (one order of magnitude for urine [48], and three orders of magnitude for serum [49]).

## 4. Conclusions

In the present study, a simple and low-cost chitosan film sensor (CFS) has been developed, characterized and tested in a gas diffusion device for detecting ammonia in biological samples. The CFS is a solvent casted self-standing film, with a thickness of 15 μm, containing the Nessler’s reagent (20% of the weight of potassium tetraiodomercurate (II)), which triggers a colorimetric reaction with molecular ammonia. The CFS has been studied in regard to its chemico-physical properties to confirm that a composite material was prepared. The SEM–EDX analysis suggests the homogeneous dispersion of Nessler’s reagent into the chitosan matrix, and shows visible formation of salt residues, which does not significantly impact the film’s transparency. The FTIR analysis shows increased intensities that could be related to differences in the chitosan coordination and in the neutralization chitosan amine groups due to the addition of Nessler’s reagent ions. The CFS thermal properties assessed by TGA and DSC confirmed the formation of a composite material. The CFS is transparent, enabling a facile evaluation of the color change, either using smartphone image analysis or the naked eye. When tested in a miniaturized gas diffusion device, the CFS can detect the presence of ammonia in samples of serum and urine, with a sensitivity suitable for application as a point-of-care analytical tool. The gas diffusion device testing setup has an extremely easy functionality for on-site analysis, since the sample can be allocated directly on the appropriately prepared gas diffusion device and the color change inspected with a phone camera.

As such, from this perspective the device, after the necessary engineering and validation, can be extremely useful in developing countries and in locations with limited access to clinical laboratories because of its portability, handiness and fast detection.

## Figures and Tables

**Figure 1 polymers-15-04238-f001:**
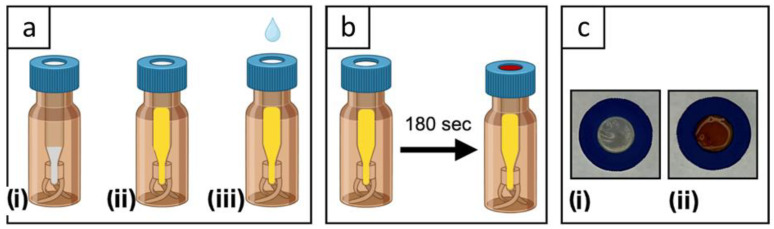
Representation of the procedure for detecting ammonia through the gas diffusion device carrying the CFS in the lid. (**a**) Steps of the procedure: (**i**) addition of 30 mg of NaOH into a vial equipped with CFS in the lid; (**ii**) addition of 200 µL of the sample into the vial; (**iii**) addition of 10 µL of NaOH 6M on the CFS. (**b**) Reaction between ammonia and Nessler’s reagent resulting in a colored CFS. (**c**) Image of the lid with CFS, sensing part of the gas diffusion device: (**i**) blank sample; (**ii**) spiked sample with ammonium.

**Figure 2 polymers-15-04238-f002:**
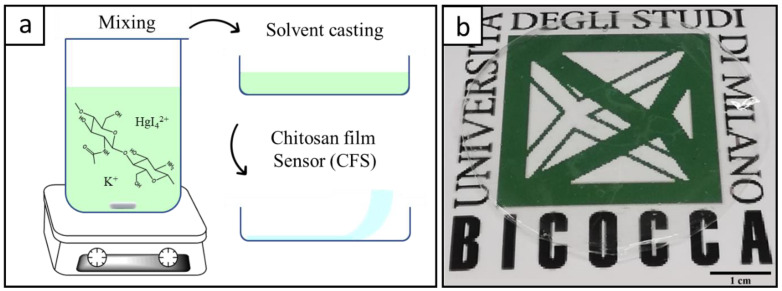
(**a**) Schematic representation of the CFS preparation: mixing of chitosan and Nessler’s reagent, solvent casting, peeling of the self-standing film; (**b**) transparent CFS film.

**Figure 3 polymers-15-04238-f003:**
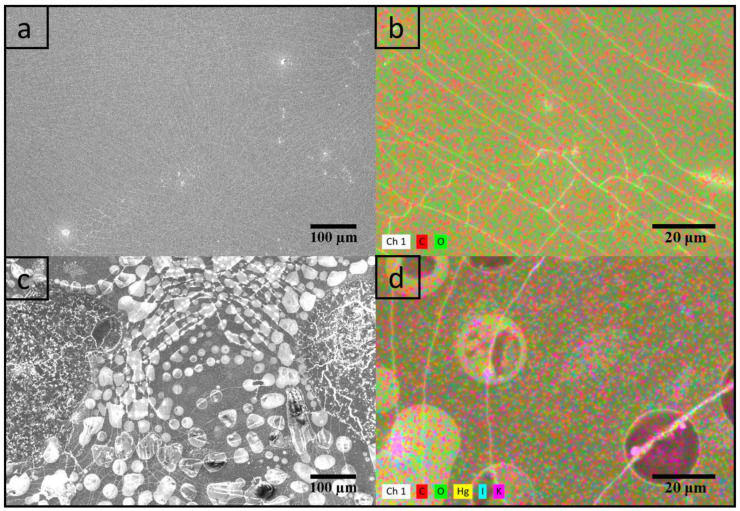
SEM and EDX analysis of a pure chitosan film (**a**,**b**) and of CFS (**c**,**d**).

**Figure 4 polymers-15-04238-f004:**
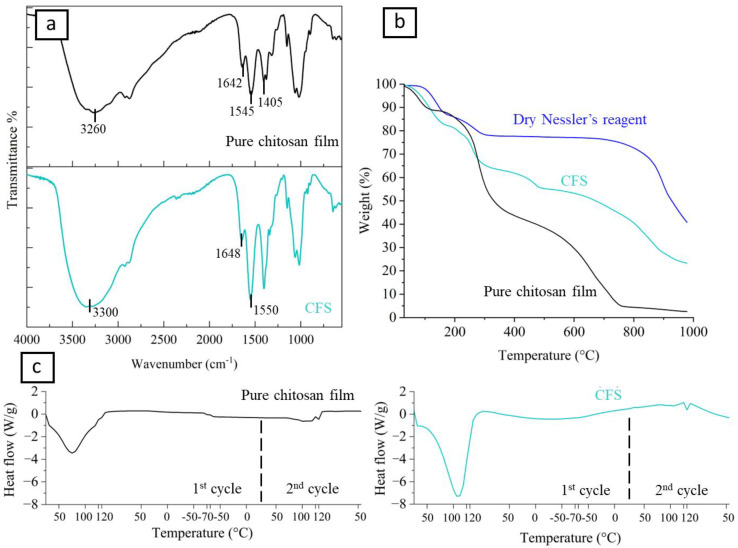
(**a**) FTIR analysis of pure chitosan film (black line) and CFS (light blue line); (**b**) TGA analysis of pure chitosan film (black line), CFS (light blue line) and dried Nessler’s reagent (blue line); (**c**) DSC analysis of pure chitosan film (black line) and CFS (light blue line).

**Figure 5 polymers-15-04238-f005:**
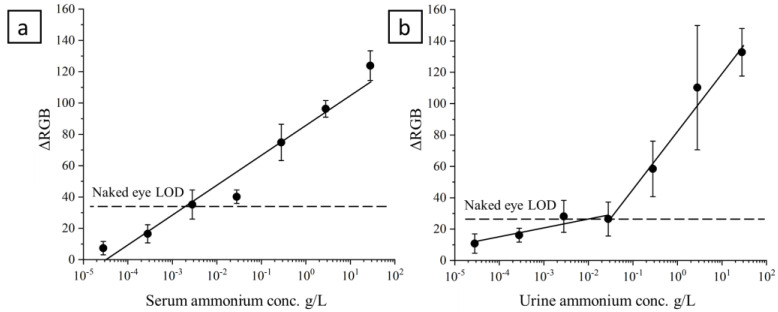
Relationship between the serum ammonium concentration (*x*-axis) and the *RGB* distance (*y*-axis). The ammonium has been spiked in the range from 28 µg/L to 28 g/L (with dilution steps equal to 10): (**a**) serum sample: y (*RGB* distance) = 19.6 (±2.3) × log10 [Serum ammonium concentration] − 86.7 (±5.9); (**b**) urine sample: y (*RGB* distance) = 5.9 (±4.0) × log10 [Urine ammonium concentration] + 38.4 (±12.9); y (*RGB* distance) = 37.1 (±8.3) × log10 [Urine ammonium concentration] + 83.9 (±9.3).

## Data Availability

The data presented in this study are available on request from the corresponding author.

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
