# Peer review of "Chitosan Film Sensor for Ammonia Detection in Microdiffusion Analytical Devices"

_polymers, 2023, doi:10.3390/polym15214238_

Round 1

Reviewer 1 Report

Comments and Suggestions for Authors

The manuscript is very interesting and important for the development of novel sensing systems.
However, the discussion must be improved. All suggestions are shown in a PDF file (marked in text with comments added).

Comments on the Quality of English Language

.

Reviewer 2 Report

Comments and Suggestions for Authors

Irene Tagliaro and coauthors developed a chitosan film sensor for ammonia detection for bio examples. The sensors were carefully characterized and tested. This work could be accepted after addressing the following issues:

1.      The introduction for chitosan films and Nessler’s reagent is not well balanced. The details on lines 64-52 seem wordy and have nothing to do with the subject.

2.      Data need to be presented for the description on lines 189-191

3.      For XDS in Figure 3, only characteristic elements should be kept. Please remove C and O.

4.      The characterization in 3.2. Physico-chemical characterization of the CFS needs to be organized or rephrased. Currently, they are like an experiment report. The reasons why you do them and the conclusion need to be properly presented.

5.      Data, calculation details, and standards for the conclusions are needed for Line 269-270

6.      Where this sensor is supposed to be used? For a sensor with solution, is it need proper encapsulation?

7.      Biocompatibility seems one of the merits of chitosan, but Nessler’s reagent is toxic for Hg? Moreover, this testing setup seems hard for in situ detection?

8.      The liquid-gas interface for gas diffusion is generally unstable to external disturbance. How to deal with it?

Round 2

Reviewer 1 Report

Comments and Suggestions for Authors

I have familiarized myself with all corrections and suggest the approval of the manuscript in its current form.

Comments on the Quality of English Language

-

Reviewer 2 Report

Comments and Suggestions for Authors

The authors have addressed all my concerns. The manuscript can be published as it is.